# Study of the Nasal Cavity of the Cadaveric Yellow-Legged Gull (*Larus michahellis atlantis*) Through Anatomical Cross-Sections and Computed Tomography

**DOI:** 10.3390/ani15213114

**Published:** 2025-10-27

**Authors:** Jose Raduan Jaber, Manuel Morales, Alvaro Ros, Pablo Paz-Oliva, Natalia Roldán-Medina, Alejandro Morales-Espino, Alberto Arencibia, Soraya Déniz

**Affiliations:** 1Departamento de Morfología, Facultad de Veterinaria, Universidad de Las Palmas de Gran Canaria, Trasmontaña, Arucas, 35413 Las Palmas, Spain; alvaro.ros101@alu.ulpgc.es (A.R.); pablo.paz101@alu.ulpgc.es (P.P.-O.); natalia.roldan101@alu.ulpgc.es (N.R.-M.); alejandro.morales108@alu.ulpgc.es (A.M.-E.); alberto.arencibia@ulpgc.es (A.A.); 2VETFUN, Educational Innovation Group, University of Las Palmas de Gran Canaria, 35413 Las Palmas, Spain; 3Grupo de Investigación en Anatomía Aplicada y Herpetopatología, University of Las Palmas de Gran Canaria, 35413 Las Palmas, Spain; 4Hospital Clínico Veterinario, Facultad de Veterinaria, Universidad de Las Palmas de Gran Canaria, Trasmontaña, Arucas, 35413 Las Palmas, Spain; manuel.morales@ulpgc.es (M.M.); soraya.deniz@ulpgc.es (S.D.)

**Keywords:** anatomy, computed tomography, gull, nasal cavity, seabird, cross-sectional anatomy

## Abstract

**Simple Summary:**

Modern imaging techniques, such as computed tomography (CT), have become valuable tools for veterinarians and researchers studying avian anatomy and disease. This study used CT and anatomical cross-sections to describe the nasal cavity and sinuses of the yellow-legged gull (*Larus michahellis atlantis*), a seabird commonly found across Europe and North Africa. By comparing CT images with anatomical sections, the main nasal structures, including the rostral, middle, and caudal nasal conchae, the nasal septum, and the infraorbital sinus, were clearly identified. This combined approach allowed accurate visualization of internal nasal features that are difficult to assess by dissection alone. The findings provide a detailed anatomical reference that can assist in diagnosing nasal and sinus diseases in gulls and other seabirds.

**Abstract:**

Understanding the anatomy of the avian nasal cavity and paranasal sinuses is essential for diagnosing respiratory diseases and interpreting imaging findings. However, detailed tomographic descriptions of these structures are scarce in seabirds. This study aimed to provide an anatomical and radiological characterization of the nasal cavity and associated sinuses of the yellow-legged gull (*Larus michahellis atlantis*). Computed tomography (CT) was performed on eight cadaveric specimens using a 16-slice helical scanner with bone and pulmonary window settings. Anatomical cross-sections of the same heads were subsequently obtained to correlate and validate CT findings. CT imaging clearly delineated major nasal structures, including the rostral, middle, and caudal nasal conchae, the nasal septum, and the infraorbital sinus, as well as their connections to adjacent cranial bones. The integration of CT and anatomical cross-sections provided detailed spatial relationships and accurate visualization of the internal nasal architecture. This study demonstrates the value of CT for examining avian cranial anatomy and provides a morphological reference framework that may aid in diagnosing nasal and sinus pathologies in seabirds.

## 1. Introduction

The yellow-legged gull (*Larus michahellis atlantis*) is a seabird belonging to the family Laridae. It is a migratory species widely distributed across Europe and Africa [1,2]. This species is primarily diurnal, although it can be observed commuting to its different feeding sites [2], where it feeds on garbage, fish, crustaceans, other birds, and carrion [1,2]. To support this dietary versatility, it possesses a specialized beak with a slightly hooked tip. It exhibits a characteristic intense yellowish orange colour, with a red spot on the mandible visible in adults, which functions as a visual stimulus to encourage feeding behaviors in chicks [1,2,3,4]. This feature is common among other gull species [2,3] and is associated with lateral nostrils positioned on the anterior portion of the beak. These nostrils are longitudinal and linear in shape, widening anteriorly [5,6]. This anatomical configuration facilitates the excretion of salt through specialized glands, a crucial adaptation for osmoregulation in marine environments [7].

Despite the ecological importance and broad distribution of gulls, detailed anatomical information regarding their nasal cavity and associated structures remains extremely limited. Most existing descriptions of the avian upper respiratory system have focused on other seabirds or terrestrial species [5,6,7], leaving a clear knowledge gap concerning gull-specific morphology.

Maintaining optimal respiratory and nasal health is crucial for the survival and well-being of gulls. However, seabirds are susceptible to various conditions affecting the nasal and respiratory disorders, including parasitic infestations (*Reighardia sternae*, *Cyathostoma lari*), infections caused by Influenza A virus, *Mycoplasma* spp., and other pathogens [8,9,10,11,12,13]. Given these vulnerabilities, veterinarians, biologists, and researchers need to expand their understanding of these species’ anatomy to gain deeper insights into their biology and the detection of potential pathologies to facilitate clinical interventions. Although some studies have investigated ocular and hepatic anatomy using sonographic and microscopic techniques [14,15], comprehensive anatomical data—particularly of the upper respiratory system—remain scarce. In this context, advanced imaging modalities offer a vital role in identifying both normal anatomical structures and associated pathological conditions.

Traditionally, conventional radiography has been the primary modality for investigating the nasal cavity in animals [16,17,18,19]. However, its diagnostic capacity is limited by the superimposition of overlapping anatomical structures, which hampers accurate visualization. In recent years, advances in diagnostic imaging have positioned computed tomography (CT) and magnetic resonance imaging (MRI) as more precise and reliable techniques for studying avian anatomy and diagnosing related pathologies [20,21,22,23,24,25].

Recent technological developments have markedly improved the application of CT in avian medicine. This imaging modality allows detailed, non-invasive visualization of complex cranial and soft tissues structures, enabling accurate three-dimensional reconstruction of the nasal cavity, sinuses, and adjacent bones. Compared with traditional radiography or dissection, CT offers superior spatial resolution and multiplanar imaging capabilities, which are particularly useful for assessing delicate anatomical regions in birds [20,21,22,23,24,25]. Its diagnostic value has also been demonstrated in the evaluation of sinusitis, trauma, and neoplastic or inflammatory processes affecting the upper respiratory tract of various avian species [20,24,26,27]. These advances establish CT as an indispensable tool for advancing both anatomical knowledge and clinical practice in avian medicine.

Despite these advantages, the application of CT to assess nasal cavity anatomy in birds remains underrepresented in the veterinary literature. Although a few studies have employed CT to describe nasal and head anatomy on species such as ostriches, parrots, shearwaters, and penguins [26,27,28,29,30], current research in this area is still limited. To the best of the authors’ knowledge, no previous studies have combined CT imaging with anatomical sections to provide a detailed examination of the nasal cavity in gulls. Therefore, the present study aimed to describe the nasal cavity of the yellow-legged gull through the integration of anatomical cross-sections and CT imaging. This combined approach provides complementary insights that can serve as a reference and anatomical atlas for gulls, supporting comparative studies with other species, enhancing the teaching of avian anatomy, improving CT interpretation, and guiding clinical diagnosis and treatment in veterinary and zoological practice.

## 2. Materials and Methods

### 2.1. Specimens

For this study, carcasses of eight subadult yellow-legged gulls (*Larus michahellis atlantis*) were provided by the Consejería de Área de Medio Ambiente, Clima, Energía y Conocimiento of the Cabildo Insular de Gran Canaria. Only carcasses classified as “*fresh dead*”(within 12 h post-mortem) and showing no external signs of decomposition (absence of bloating, discoloration, or feather loss) were included. The specimens had an average body weight of 0.7451 kg (ranging from 0.6319 to 0.9574 kg), with a mean skull length of 12.537 cm measured from the beak to the back of the head (range from 11.3 to 14.5 cm), and a beak length of 7.66 cm (range from 7.3 to 8 cm). Table 1 summarizes the morphometric data (body weight, head length, and beak length) obtained from all individuals. These measurements were recorded to document interindividual variability among specimens. However, no internal morphometric measurements were performed, as the limited sample size precluded establishing reliable reference ranges. All birds included in the study were either found deceased or in a debilitated state and subsequently died despite rescue efforts. Following collection, all specimens were sealed in moisture-proof bags and stored at −80 °C until imaging and sectioning. No animals were euthanized for this study.

### 2.2. CT Technique

For CT, frozen heads were transferred to 4 °C and thawed until free of surface frost and motionless positioning could be achieved. We conducted transverse and sagittal CT scans with a 16-slice helical CT scanner (Toshiba Astelion, Canon Medical System, Tokyo, Japan). The birds were placed in symmetrical dorsal recumbency on the examination table, with craniocaudal beam orientation. Scanning was conducted under a standardized protocol with the following acquisition parameters: 120 kVp, 80 mA, 512 × 512 acquisition matrix, 1809 × 858 field of view, a pitch of 0.94, and a gantry rotation time of 1.5 s. These images had a slice thickness of 0.6 mm. To improve the identification of the anatomical formations on CT scans, we used different CT window settings by adjusting the window widths (WWs) and window levels (WLs): a bone window setting (WW = 1500; WL = 300) and a pulmonary window setting (WW = 1400; WL = −500). This last window was intentionally selected for its ability to enhance contrast between air-filled spaces and the surrounding thin bony and soft tissue structures. No conspicuous interindividual variability was detected among the examined specimens. The general configuration and relative proportions of the nasal conchae, nasal septum, and infraorbital sinus remained consistent across all individuals, with only minor differences in the curvature of the rostral nasal concha and the extent of infraorbital sinus pneumatization, which were considered within the normal range of individual variation. Additionally, we utilized the original data to generate volume-rendered reconstructed images using a standard Dicom 3D format (OsiriX MD, Geneva, Switzerland).

### 2.3. Anatomical Sections

Following completion of the imaging procedures, all the specimens were positioned in dorsal decubitus within expanded polystyrene containers and refrozen at −80 °C for 72 h. After freezing, serial anatomical sections—each 0.8 cm thick—were produced using an electric band saw to obtain transverse and sagittal views. The slices were then promptly rinsed with water to eliminate any artefacts, labelled sequentially, and photographed on both sides. To facilitate precise localization of anatomic structures, we performed a targeted dissection that linked each anatomic cross-section to its CT representation. This step markedly improved the accuracy of identification and the fidelity of anatomic–radiologic correlation. However, it should be noted that the 0.8 cm thickness of the anatomical sections, compared with the 0.6 mm CT slice thickness, implies that the anatomical images were representative and intended for comparative orientation rather than exact 1:1 correspondence with specific CT slices.

The anatomical figures presented in this study were obtained from several individuals. For each section, the best-preserved and most representative samples were selected, primarily from larger specimens, to maximize correspondence between the anatomical sections and CT images. Owing to the small head size and high interindividual similarity, anatomical alignment across specimens was feasible and consistent.

### 2.4. Anatomical Evaluation

The anatomical cross-sections that better matched the CT images were selected to facilitate the identification of relevant structures within the nasal cavity of the yellow-legged gull. Additionally, reference materials, including avian anatomy textbooks, peer-reviewed literature, and anatomical preparations from other seabird specimens, were consulted to enhance anatomical interpretation and ensure accurate morphological comparisons [20,21,22,23,24,25,26,27,28,29,30,31].

## 3. Results

A series of figures illustrating the anatomical features of the nasal cavity and related structures in the yellow-legged gull are presented (Figure 1, Figure 2, Figure 3, Figure 4, Figure 5, Figure 6, Figure 7, Figure 8, Figure 9, Figure 10, Figure 11 and Figure 12). Figure 1 provides a lateral view of the head, with each numbered line indicating the approximate anatomical levels corresponding to the transverse anatomical and CT sections shown in Figure 4, Figure 5, Figure 6, Figure 7, Figure 8, Figure 9, Figure 10, Figure 11 and Figure 12. Each of these figures comprises three image types: (A) a transverse anatomical cross-section, (B) a transverse CT scan with a pulmonary window setting, and (C) a transverse CT scan with a bone window setting. The series follows a rostro-caudal progression, beginning at the beak and extending toward the orbital fossa, except for Figure 2 and Figure 3, which correspond to paramedian sections. Finally, Figure 12 shows (A,B) a three-dimensional CT reconstruction and (C,D) an Osirix mask CT reconstruction of the yellow-legged gull’s head in lateral view.

### 3.1. Anatomical Cross-Sections

Clinically pertinent anatomy of the nasal cavity was delineated on detailed cross-sectional images. The selected levels illustrate the principal landmarks along the rostrocaudal axis—from the external nares to the choanal slit. Consequently, the nostrils were observed rostrally with an elongated shape preventing the entrance of excessive water into the nasal cavity (Figure 4A). Caudally to the nares, the nasal cavity was separated by the median nasal septum (Figure 5A, Figure 6A, Figure 7A and Figure 8A). Hence, both nostrils and the nasal cavity were bounded by the premaxilla (Figure 3A and Figure 4A), maxilla (Figure 4A, Figure 5A, Figure 6A, Figure 7A and Figure 8A), nasal (Figure 2A, Figure 3A, Figure 5A, Figure 6A, Figure 7A and Figure 8A), palatine (Figure 5A, Figure 6A, Figure 7A, Figure 8A and Figure 9A), frontal (Figure 2A, Figure 3A, Figure 9A, Figure 10A and Figure 11A) and lacrimal (Figure 10A) bones. Ventrally to the nasal septum and in a medial position all along the nasal cavity, we displayed the vomer (Figure 5A, Figure 6A, Figure 7A, Figure 8A, Figure 9A, Figure 10A and Figure 11A).

Furthermore, among both sides of the nasal septum the nasal conchae were clearly distinguished showing the rostral nasal concha in a dorsolateral position in the nasal cavity (Figure 3A, Figure 5A, Figure 6A and Figure 7A), the middle nasal concha was located ventrolaterally and shifted to a medial position caudally (Figure 2A, Figure 3A, Figure 6A, Figure 7A, Figure 8A and Figure 9A), and the caudal nasal concha dorsally to the infraorbital sinus (Figure 2A, Figure 3A, Figure 9A and Figure 10A). By a considerable size, the middle nasal concha was the biggest and presented spiral lamellae. Additionally, a sinus that may be relevant to flight was found rostroventrally to the eye, corresponding to the infraorbital sinus (Figure 2A, Figure 5A, Figure 6A, Figure 7A, Figure 8A, Figure 9A, Figure 10A and Figure 11A). In addition, the nasal meatuses (Figure 2A, Figure 3A, Figure 8A, Figure 10A and Figure 11A) were clearly identified in the sagittal and transverse sections.

Meanwhile, more caudal sections allowed the distinction of an aperture connecting the nasal cavity to the oral cavity called the choanal cleft. Moreover, a clear visualization of the oral cavity was achieved in all the transverse anatomical sections (Figure 2A, Figure 3A, Figure 4A, Figure 5A, Figure 6A, Figure 7A, Figure 8A, Figure 9A, Figure 10A and Figure 11A). This cavity hosted the large tongue (Figure 4A, Figure 5A, Figure 7A, Figure 8A and Figure 9A) and was surrounded by the mandible (Figure 2A, Figure 3A, Figure 4A, Figure 5A, Figure 6A, Figure 7A, Figure 8A, Figure 9A, Figure 10A and Figure 11A) and different mandibular muscles (Figure 2A, Figure 6A, Figure 7A, Figure 8A, Figure 9A, Figure 10A and Figure 11A). In addition, specific components of the hyobranchial apparatus were also identifiable. Notably the paraglossum and the ceratobranchiale, which play a critical role in supporting and mobilizing the tongue (Figure 9A, Figure 10A and Figure 11A).

### 3.2. Computed Tomography (CT)

CT scans closely mirror the gross anatomy and allow confident structure-by-structure correlation. This procedure displayed several components of the nasal cavity, such as the nostrils (Figure 3B and Figure 4B,C), which were represented as an air-density structure on the lateral sides of the beak. Other relevant structures, such as the nasal septum (Figure 5B, Figure 6B, Figure 7B and Figure 8B), was visible as a thin soft-tissue band with the pulmonary window. This same feature happened in structures like the nasal conchae. Therefore, the rostral nasal concha (Figure 2B, Figure 3B, Figure 5B and Figure 6B), was depicted as a hypoattenuated structure found in the lateral and rostral part of the nasal cavity; the middle nasal concha, which was clearly identified in the pulmonary window (Figure 2B, Figure 3B, Figure 6B, Figure 7B, Figure 8B and Figure 9B) on one of the sides, appeared as an hypoattenuated structure, and becomes more hyperattenuated caudally; and the caudal nasal concha (Figure 2B, Figure 3B and Figure 10B) was displayed as hypoattenuated structure that shifts from a dorsal to a more ventral location. Other associated structures were the nasal meatus (Figure 8B,C, Figure 10B,C and Figure 11B,C), the frontal sinus (Figure 10B and Figure 11B), and the infraorbital sinus (Figure 5B, Figure 6B, Figure 7B, Figure 8B, Figure 9B, Figure 10B,C and Figure 11B,C), all of which were presented as hypoattenuated and were mostly found in the pulmonary window.

The sagittal and transverse computed tomography images also enabled clear visualization in pulmonary and bone windows of various cranial bones, including the premaxilla (Subfigures B and C in Figure 2, Figure 3 and Figure 4), maxilla (Figure 2B and Subfigures B and C in Figure 4, Figure 5, Figure 6, Figure 7 and Figure 8), nasal (Subfigures B and C in Figure 2, Figure 3, Figure 5, Figure 6, Figure 7 and Figure 8), palatine (Figure 5B and Subfigures B and C in Figure 6, Figure 7, Figure 8 and Figure 9), frontal (Subfigures B and C in Figure 2, Figure 3, Figure 9, Figure 10 and Figure 11), and vomer (Figure 5B, Figure 6B, Figure 7B,C, Figure 8B,C, Figure 9B,C and Figure 10B,C and Figure 11B). In addition to these skeletal elements, various elements of the hyobranchial apparatus, including the paraglossum and the ceratobranchiale, which were hyperattenuated in both windows (Subfigures B and C in Figure 8, Figure 9, Figure 10 and Figure 11).

No conspicuous interindividual variability was detected among the examined specimens. The general configuration and relative proportions of the nasal conchae, nasal septum, and infraorbital sinus remained consistent across all individuals, with only minor differences in the curvature of the rostral nasal concha and the extent of infraorbital sinus pneumatization, which were considered within the normal range of individual variation.

Volume-rendered and mask-based three-dimensional (3D) reconstructions complemented the sectional anatomy by depicting the conchal arrangement, sinuses, and orbital relationships in lateral view (Figure 12A–D). In addition, these reconstructions offered an alternative perspective and allowed a more comprehensive assessment and deeper understanding of the cranial bones identified. Moreover, other structures such as the eye, the scleral ring, oral cavity, and pharynx (Figure 12A–D) were clearly distinguishable.

## 4. Discussion

Gulls inhabit dynamic marine environments characterized by variable thermal and barometric conditions. Therefore, a detailed understanding of their cranial anatomy is pivotal in elucidating adaptations related to thermoregulation, pressure equalization, and respiratory efficiency.

Consistent with broader patterns in Laridae, our gross and cross-sectional examinations delineated the bony boundaries of the nasal cavity. In gulls, this cavity is fully divided into right and left passages by the median nasal septum. This septum, in line with general avian anatomy, comprises a rostral cartilaginous plate and a caudal osseous component [30,31]. By contrast, some waterfowl (ducks and Wanxi white geese) exhibit an elongated rostral perforation that allows communication between nasal chambers—an interspecific difference within aquatic birds not observed in our specimens [32]. CT-based stereological work in *Larus fuscus* corroborates the presence and relative proportions of the nasal septum and conchae and supports CT as a reproducible approach for volumetric assessment and study of the nasal architecture [33]; similar methodology was validated in domestic geese [26].

Within the nasal cavity of the yellow-legged gull (*Larus michahellis atlantis*), we could distinguish three distinct nasal conchae—rostral, middle, and caudal—each contributing to increased surface area and enhanced airflow regulation. This tripartite conchal arrangement aligns with findings in other seabirds such as Cory’s shearwater and certain waterfowl species, including ducks, geese, and turkeys [20,34,35,36]. The anatomic sagittal sections and CT images obtained with pulmonary window settings displayed excellent detail of the large development of the middle nasal concha and its spiral configuration, suggesting a potential role in thermoregulation and particle filtration. In contrast, species such as the hooded crow and dove possess only two nasal conchae [37,38], whereas parrots have been reported to lack nasal conchae entirely [39]. These interspecific differences highlight the adaptive diversity of avian nasal architecture with respect to ecological niche and respiratory demands. The pronounced development of the middle concha and the extensive infraorbital sinus observed in this species may represent adaptive features to the marine environment, contributing to efficient air humidification and thermoregulation, while also assisting in the management of saline exposure associated with seabird physiology.

The infraorbital (antorbital) sinus differs markedly from mammalian paranasal sinuses. It lies lateral to the nasal cavity, extends rostrally toward the beak and ventrally around the globe, and communicates dorsomedially with the nasal conchae and caudallywith the cervicocephalic air sac. This configuration—characterized by thin mucosa, multiple diverticula, and limited drainage routes—has both functional and clinical relevance, particularly in the context of inflammation or infection [39,40,41,42]. The sinus’s extensive pneumaticity contributes to cranial lightening while maintaining structural strength, a trait advantageous in flying species. Moreover, its integration with the continuous avian respiratory system may assist in equilibrating pressure changes and modulating airflow within the skull during flight.

In this study, multiplanar CT reconstructions with different window settings proved especially valuable for delineating this sinus and tracing its extension from the rostral nasal concha to the ventral aspect of the eyeball. Our findings align with those of Mohamed (2008) [43], who reported both infraorbital and frontal sinuses in the ostrich, whereas later studies documented only the frontal sinus [31,44]. Methodological differences—particularly the use of dissection or plain radiography instead of cross-sectional imaging—likely explain this discrepancy. As in other seabirds such as Cory’s shearwater, the infraorbital sinus exhibited a triangular configuration, contrasting with the rounder shape observed in the moorhen and broad-breasted white turkey [45]. Although its precise physiological role remains to be determined, recent CT-based studies in seabirds, supported by computational airflow models in other avian taxa, suggest that this complex air space could contribute to cranial pneumatic regulation and thermoregulatory efficiency during flight [20,26,42,46].

This study has several limitations. First, imaging and sectioning were performed on cadaveric specimens that were frozen prior to processing; post-mortem change and freeze–thaw cycles may subtly alter mucosal thickness and air–soft tissue interfaces. Second, the mismatch between CT slice thickness (0.6 mm) and gross section thickness (8 mm) introduces partial volume effects and complicates one-to-one registration despite careful landmarking. Third, the sample size is modest and lacks histological confirmation, which precludes fine tissue-level attribution (e.g., cartilage vs. thin bone in the septum) in ambiguous regions. Fourth, our conclusions are anatomical; we did not obtain in vivo functional data (e.g., airflow, pressure, or temperature) and therefore cannot infer physiology beyond plausible hypotheses. Finally, detectability of thin structures is window-dependent and may vary with scanner, kernel, and voxel size; reproducibility across platforms warrants future validation. Other imaging modalities, notably microcomputed tomography (micro-CT), can achieve substantially higher spatial resolution and thinner reconstructed slices than conventional clinical CT. Nevertheless, limited gantry aperture, longer acquisition times, and restricted availability mean that micro-CT systems are seldom installed in veterinary hospitals, limiting routine clinical use [47].

## 5. Conclusions

This investigation provides a detailed anatomical and radiological characterization of the nasal cavity and paranasal sinuses in the yellow-legged gull (*Larus michahellis atlantis*) through the integration of computed tomography and anatomical cross-sections. The combined approach enabled precise identification of key nasal structures, including the rostral, middle, and caudal nasal conchae, the infraorbital sinus, and associated bones such as the premaxilla, maxilla, vomer, and frontal. The use of different CT window settings enhanced the visualization of both soft tissue and air-filled cavities, demonstrating the utility of CT imaging in avian anatomical studies.

These findings offer a valuable reference for clinicians, anatomists, and researchers working with seabirds, particularly in the context of diagnosing nasal pathologies and planning surgical or rehabilitative interventions. Moreover, the anatomical atlas generated from this work contributes to the broader understanding of avian cranial morphology and supports comparative studies across species. Future research could expand on these results by incorporating functional imaging or histological analysis to further elucidate the physiological roles of the nasal structures described. Additionally, the incorporation of plastination or dry skull preparations could further complement and expand upon the present findings, enhancing the three-dimensional understanding of osteological and soft-tissue relationships within the gull’s nasal cavity.

## Figures and Tables

**Figure 1 animals-15-03114-f001:**
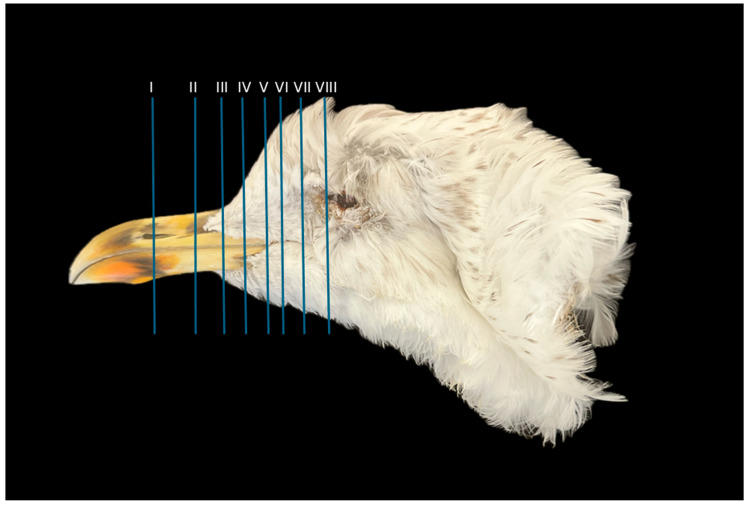
Approximate anatomical level sections of the of the yellow-legged gull’s nasal cavity. Lines depict the transverse (I to VIII) images.

**Figure 2 animals-15-03114-f002:**
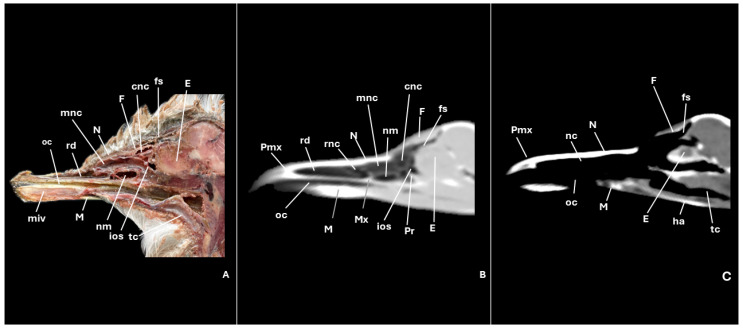
Specimen 7. Parasagittal section (**A**), CT pulmonary window (**B**) and CT bone window (**C**) images showing the yellow-legged gull’s nasal cavity. cnc: caudal nasal concha; E: eye; F: frontal bone; fs: frontal sinus; ha: hyobranchial apparatus; ios: infraorbital sinus; M: mandible; miv: ventral intermandibular muscle; mnc: middle nasal concha; N: nasal bone; nc: nasal cavity; nm: nasal meatus; oc: oral cavity; Pmx: premaxillary bone; Pr: parasphenoid rostrum; rd: rostral diverticulum; rnc: rostral nasal concha; tc: trachea.

**Figure 3 animals-15-03114-f003:**
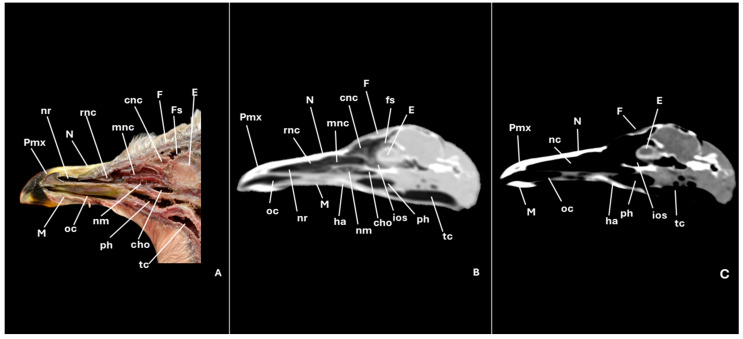
Specimen 1. Parasagittal section (**A**), CT pulmonary window (**B**) and CT bone window (**C**) images illustrating the yellow-legged gull’s nasal cavity. cho: choanal opening; cnc: caudal nasal concha; E: eye; F: frontal bone; Fs: frontal sinus; ha: hyobranchial apparatus; ios: infraorbital sinus; M: mandible; mnc: middle nasal concha; N: nasal bone; nc: nasal cavity; nm: nasal meatus; nr: nostrils; oc: oral cavity; ph: pharynx; Pmx: premaxillary bone; rnc: rostral nasal concha; tc: trachea.

**Figure 4 animals-15-03114-f004:**
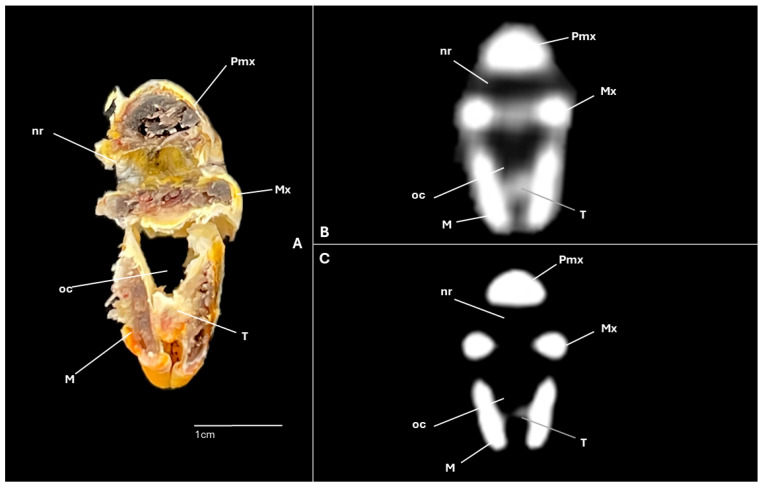
Specimen 1. Transverse cross-section (**A**), CT pulmonary window (**B**) and CT bone window (**C**) images of the yellow-legged gull’s nasal cavity corresponding to line I in Figure 1. M: mandible; Mx: maxilla; nr: nostril; oc: oral cavity; Pmx: premaxillary bone; T: tongue.

**Figure 5 animals-15-03114-f005:**
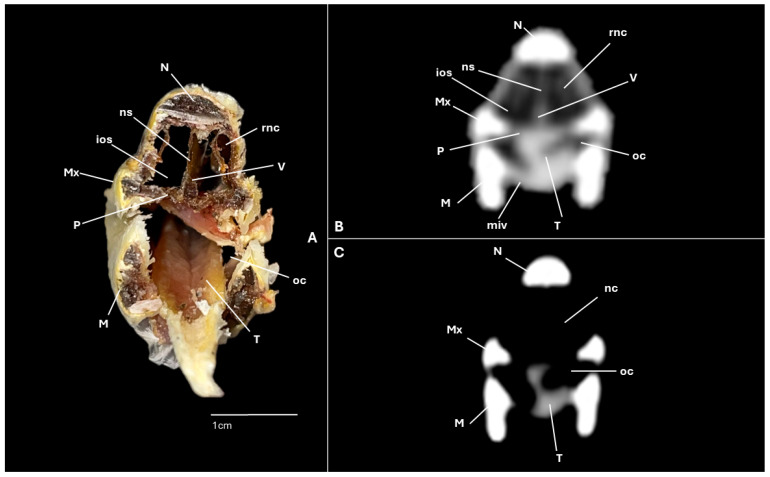
Specimen 4. Transverse cross-section (**A**), CT pulmonary window (**B**) and CT bone window (**C**) images of the yellow-legged gull’s nasal cavity corresponding to line II in Figure 1. ios: infraorbital sinus; M: mandible; miv: ventral intermandibular muscle; Mx: maxilla; N: nasal bone; nc: nasal cavity; ns: nasal septum; oc: oral cavity; P: palatine bone; rnc: rostral nasal concha; T: tongue; V: vomer.

**Figure 6 animals-15-03114-f006:**
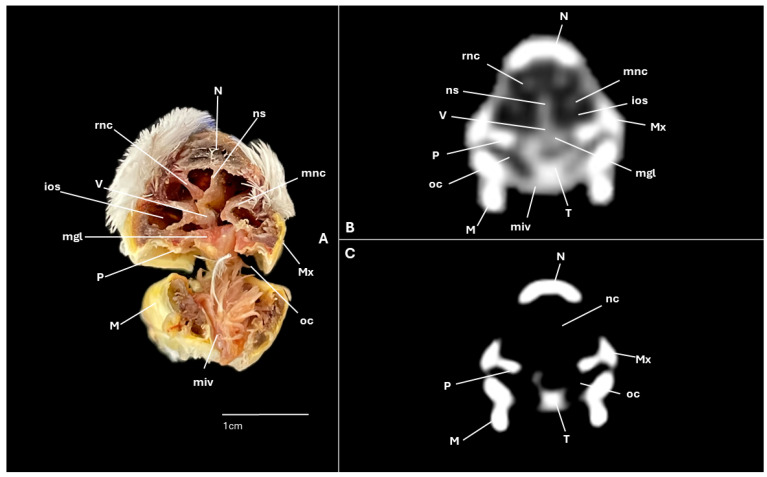
Specimen 3.Transverse cross-section (**A**), CT pulmonary window (**B**) and CT bone window (**C**) images of the yellow-legged gull’s nasal cavity corresponding to line III in Figure 1. ios: infraorbital sinus; M: mandible; mgl: maxillary salivary gland; mnc: middle nasal concha; miv: ventral intermandibular muscle; Mx: maxilla; N: nasal bone; nc: nasal cavity; ns: nasal septum; oc: oral cavity; P: palatine bone; rnc: rostral nasal concha; T: tongue; V: vomer.

**Figure 7 animals-15-03114-f007:**
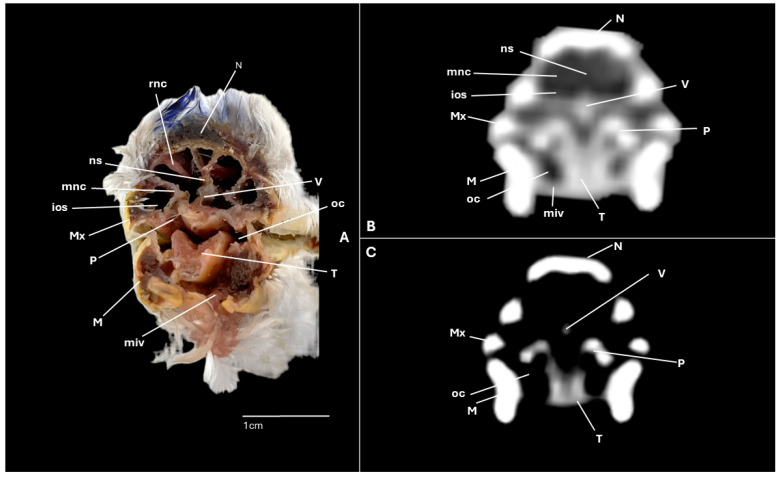
Specimen 1. Transverse cross-section (**A**), CT pulmonary window (**B**) and CT bone window (**C**) images of the yellow-legged gull’s nasal cavity corresponding to line IV in Figure 1. ios: infraorbital sinus; M: mandible; mnc: middle nasal concha; miv: ventral intermandibular muscle; N: nasal bone; ns: nasal septum; P: palatine bone; rnc: rostral nasal concha; oc: oral cavity; T: tongue; V: vomer.

**Figure 8 animals-15-03114-f008:**
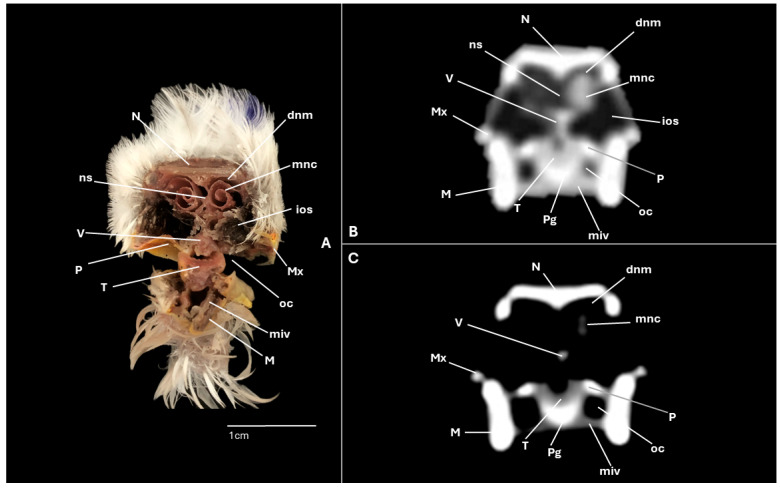
Specimen 1. Transverse cross-section (**A**), CT pulmonary window (**B**) and CT bone window (**C**) images of the yellow-legged gull’s nasal cavity corresponding to line V in Figure 1. dnm: dorsal nasal meatus; ios: infraorbital sinus; M: mandible; mnc: middle nasal concha. miv: ventral intermandibular muscle; Mx: maxilla; N: nasal bone; ns: nasal septum; oc: oral cavity; P: palatine bone; Pg: Paraglossum; T: tongue; V: vomer.

**Figure 9 animals-15-03114-f009:**
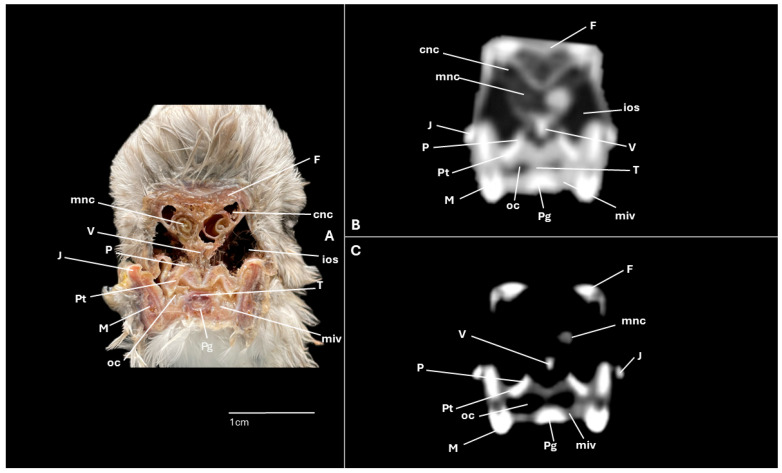
Specimen 6. Transverse cross-section (**A**), CT pulmonary window (**B**) and CT bone window (**C**) images of the yellow-legged gull’s nasal cavity corresponding to line VI in Figure 1. cnc: caudal nasal concha; F: frontal bone; ios: infraorbital sinus; J: jugal bone; M: mandible; miv: ventral intermandibular muscle; mnc: middle nasal concha; oc: oral cavity; P: palatine bone; Pg: Paraglossum; Pt: Pterygoid; T: tongue; V: vomer.

**Figure 10 animals-15-03114-f010:**
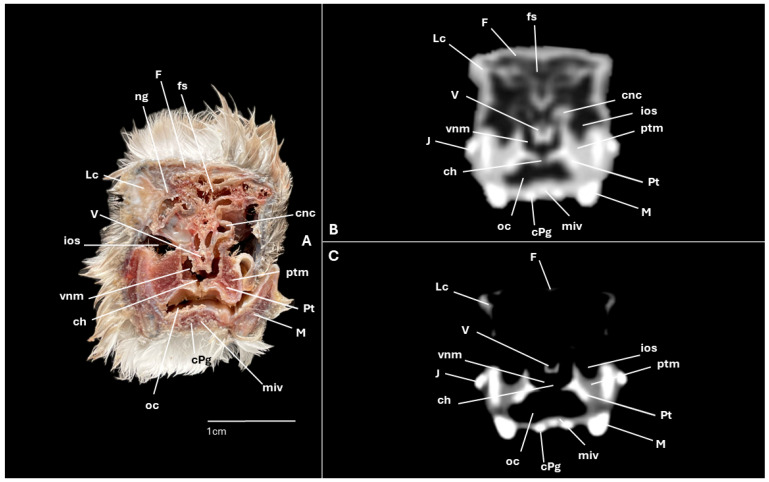
Specimen 2. Transverse cross-section (**A**), CT pulmonary window (**B**) and CT bone window (**C**) images of the yellow-legged gull’s nasal cavity corresponding to line VII in Figure 1. ch: choanal cleft; cnc: caudal nasal concha; cPg: cornu of paraglossum; F: frontal bone; fs: frontal sinus; ios: infraorbital sinus; J: jugal bone; Lc: lacrimal bone; M: mandible; miv: ventral intermandibular muscle; ng: nasal gland; oc: oral cavity; ptm: pterygoid muscle; vnm: ventral nasal meatus; V: vomer.

**Figure 11 animals-15-03114-f011:**
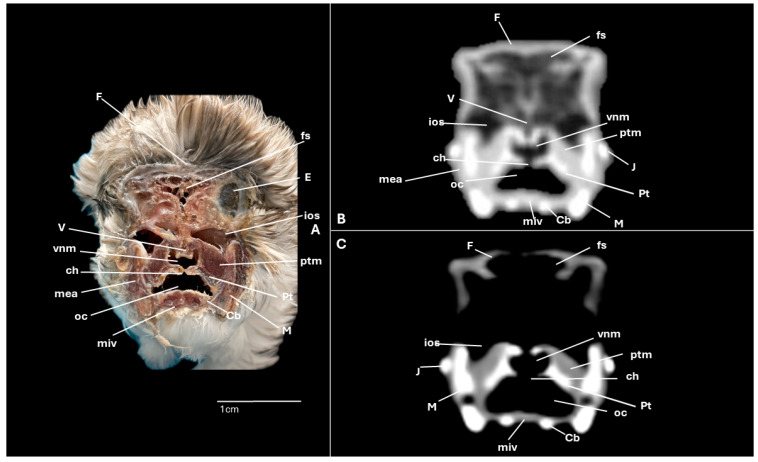
Specimen 6. Transverse cross-section (**A**), CT pulmonary window (**B**) and CT bone window (**C**) images of the yellow-legged gull’s nasal cavity corresponding to line VIII in Figure 1. Cb: ceratobranchiale; ch: choana; E: eye; F: frontal bone; fs: frontal sinus; ios: infraorbital sinus; J: jugal bone; M: mandible; mea: external adductor mandibular muscle; miv: ventral intermandibular muscle; oc: oral cavity; ptm: pterygoid muscle; V: vomer.

**Figure 12 animals-15-03114-f012:**
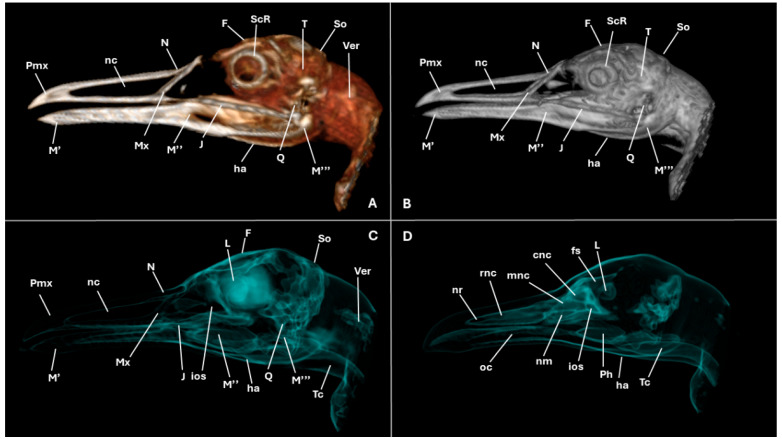
Three-dimensional (3D) CT reconstructions of the yellow-legged gull’s head. Volume-rendered 3D reconstruction (**A**,**B**), and semi-transparent 3D “mask” reconstruction isolating bone and air spaces (**C**,**D**). cnc: caudal nasal concha; F: frontal bone; fs: frontal sinus; ha: hyobranchial apparatus; ios: infraorbital sinus; J: jugal bone; L: lens; M’: mandible (dentary bone); M’’: mandible (surangular); M’’’: mandible (angular); mnc: middle nasal concha; Mx: maxilla; N: nasal bone; nc: nasal cavity; nm: nasal meatus; nr: nostrils; oc: oral cavity; Ph: pharynx; Pmx: premaxillary bone; Q: quadrate bone; rnc: rostral nasal concha; ScR: scleral ring; So: supraoccipital bone; T: temporal bone; Tc: trachea; Ver: vertebrae.

**Table 1 animals-15-03114-t001:** Weight and measurements (head and beak) of the 8 yellow-legged gulls.

Animal	Weight (g)	Head Length (cm)	Beak Length (cm)
1	957.4	14.5	8.0
2	663.3	11.5	7.5
3	776.9	12.4	7.5
4	785.1	12.4	7.8
5	631.9	13.2	7.8
6	673.5	11.3	7.4
7	807.2	12.5	8.0
8	665.5	12.5	7.3

## Data Availability

The information is available at https://accedacris.ulpgc.es (accessed on 1 October 2025).

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
