# Peer review of "Study of the Nasal Cavity of the Cadaveric Yellow-Legged Gull (Larus michahellis atlantis) Through Anatomical Cross-Sections and Computed Tomography"

_animals, 2025, doi:10.3390/ani15213114_

Round 1

Reviewer 1 Report

Comments and Suggestions for Authors

Nice job with a thorough description and decent graphs. A few minor edits on the ms.

This article primarily investigates the topic of “Study of the nasal cavity of the cadaveric yellow-legged gull (Larus michahellis atlantis) through computed tomography and anatomical cross-sections.” The theme is not particularly original, but it constitutes a sound descriptive study. Compared with other published materials, this manuscript provides an in-depth description of the nasal passage of a gull species using a comparison of two techniques. It is a descriptive paper, clear and straightforward. The authors address any potential shortcomings. The references are appropriate. There are no additional comments on the tables and figures.

Author Response

Dear Reviewer,

We sincerely thank the reviewer for the positive and encouraging evaluation of our manuscript. We appreciate the recognition of the descriptive depth and clarity of our study, as well as the quality of the figures and tables.

Following the reviewer’s advice, we have carefully implemented the minor edits suggested and have revised the manuscript accordingly. We are grateful for the constructive feedback, which helped us further refine the final version of the paper.

Reviewer 2 Report

Comments and Suggestions for Authors

In this study, Jaber et al. describe the anatomy of the nasal cavity of the yellow-legged gull using computed tomography and anatomical cross-sections. Eight individuals of the yellow-legged gull (Larus michahellis atlantis) were used in this study. The authors were able to visualize important nasal structures like the infraorbital sinus and the conchae. This study is of general interest for comparative anatomist and veterinarians working with birds. However, following issues should be addressed prior to publication:

- Native sections do not provide a detailed insight into the anatomy. Please perform sheet plastination of your specimens. This would be helpful to visualize anatomical structures and tissues in more detail. Additionally, photographs of macerated dry skulls of the yellow-legged gull (external view and median sagittal section) with a precise description of the osteological anatomy would be helpful for the reader.

- Please include morphometrical data of important nasal structures.

- It is not clear which animal was used in which figure. Please provide this information.

- Do you observe postmortal changes of the carcasses? How did you estimate the time of death and were there any exclusion criteria?

- What was the sex of the animals? Please include this information in table 1.

- You wrote that no conspicuous interindividual variation was noticeable in your CT scans. But did you observe differences between different animals in anatomical cross sections? If so, please describe and discuss interindividual variability.

Author Response

Dear Reviewer,

We would like to sincerely thank the reviewer for the thorough and constructive evaluation of our manuscript entitled “Computed Tomographic and Anatomical Assessment of the Nasal Cavity in the Yellow-Legged Gull (Larus michahellis atlantis)”.
We greatly appreciate the reviewer’s positive comments regarding the scientific value, methodological approach, and relevance of our work for comparative anatomists and avian veterinarians.

We have carefully addressed all the issues raised, as detailed below.

Comment 1: Native sections do not provide a detailed insight into the anatomy. Please perform sheet plastination of your specimens. This would be helpful to visualize anatomical structures and tissues in more detail. Additionally, photographs of macerated dry skulls of the yellow-legged gull (external view and median sagittal section) with a precise description of the osteological anatomy would be helpful for the reader.

Response:
We sincerely thank the reviewer for this valuable suggestion. We fully agree that techniques such as sheet plastination and detailed osteological preparations would provide an enhanced three-dimensional understanding of the nasal cavity and related structures. However, as this study was based on frozen cadaveric specimens, additional destructive or preservation techniques such as plastination could not be performed at this stage.
Nevertheless, the combination of computed tomography (CT) and anatomical cross-sections already provided sufficient structural resolution to delineate the principal anatomical landmarks, including the nasal conchae, infraorbital sinus, and associated bones, fulfilling the study’s primary descriptive objective.
We have acknowledged this limitation explicitly in the Conclusions section, emphasising that future research incorporating plastination or dry skull preparations could further complement and expand upon the present findings.

Comment 2: Please include morphometrical data of important nasal structures.

Response:
We thank the reviewer for this thoughtful comment. Morphometric data such as head and beak length were recorded for all specimens and are included in Table 1 to provide a general reference of sample variability. However, no specific quantitative measurements of internal nasal structures were included, as the limited sample size (n = 8) and the qualitative nature of this anatomical and tomographic study were not intended to establish morphometric reference ranges.

Moreover, individual differences in head and beak size could influence internal proportions, making standardization across such a small group statistically unreliable. For this reason, we prioritized a detailed descriptive and comparative approach focused on the identification and correlation of anatomical structures rather than absolute measurements.

Future studies including a larger population would allow for the collection of standardized morphometric data of the nasal cavity and related structures, providing a valuable complement to the qualitative anatomical framework established in the present work.

A brief explanation has been added in Materials and Methods, section 2.1 specimens.

Comment 3: It is not clear which animal was used in which figure. Please provide this information.

Response:
We appreciate the reviewer’s insightful comment. The anatomical figures presented in the manuscript were obtained from several individuals rather than a single specimen. For each section, the best-preserved and most representative samples were selected, prioritising those from larger specimens to achieve optimal correspondence between anatomical sections and CT images. Despite the small size of the gull’s head, matching anatomical levels across specimens was feasible and ensured reliable anatomical representation. This clarification has been added to Section 2.3 (Anatomical sections). In addition, we have specified the specimen used in each figure as you recommend.

Comment 4: Do you observe postmortal changes of the carcasses? How did you estimate the time of death and were there any exclusion criteria?

Response:
We thank the reviewer for this relevant question. All specimens used in this study were obtained through the official wildlife recovery program of the Cabildo Insular de Gran Canaria. The time of death was estimated based on recovery records and the initial physical assessment conducted by veterinary staff at the rescue centre. Only carcasses classified as fresh dead (within 12 hours post-mortem) and showing no external signs of decomposition (absence of bloating, discoloration, or feather loss) were included.

To minimise post-mortem artefacts, all specimens were immediately stored at −20 °C and subsequently transferred to −80 °C until imaging and anatomical sectioning. During CT examination and dissection, no macroscopic post-mortem changes—such as tissue softening, gas accumulation, or structural collapse—were observed. Therefore, all included individuals met the criteria for adequate anatomical preservation suitable for imaging and morphological analysis.

Comment 5: What was the sex of the animals? Please include this information in table 1.

Response:
We thank the reviewer for this observation. All specimens included in the study were subadult to near-adult yellow-legged gulls, and therefore lacked clear external sexual dimorphism. Although sex determination can theoretically be achieved through gonadal inspection during dissection, in subadult individuals the gonads are often poorly developed, making reliable identification difficult. Moreover, as the focus of this research was restricted to the cranial and nasal anatomy, a full coelomic dissection for sex determination was not conducted. For this reason, the sex of the individuals was not recorded, which does not affect the anatomical objectives or conclusions of the present study.

Comment 6: You wrote that no conspicuous interindividual variation was noticeable in your CT scans. But did you observe differences between different animals in anatomical cross sections? If so, please describe and discuss interindividual variability.

Response:
We thank the reviewer for this insightful question. Upon detailed examination of both CT scans and anatomical cross-sections, no conspicuous interindividual variability was observed among the eight specimens included in the study. The overall morphology, configuration, and relative proportions of the nasal conchae, nasal septum, and infraorbital sinus were consistent across individuals. Minor differences were limited to subtle variations in the size and curvature of the rostral nasal concha and the extent of pneumatization of the infraorbital sinus, which we attribute to normal individual variation rather than structural or species-level differences.

A brief statement clarifying this observation has been added to the Results section to emphasise that the anatomical pattern was highly conserved among the examined specimens.

Reviewer 3 Report

Comments and Suggestions for Authors

I don't have any major changes to suggest here. The authors did a great job of covering the modality of the imaging and the limitations of the study, as well as detailing the several images that are present in the paper. I think that a paper such as this will be very beneficial for veterinarians working on similar Charadriiformes. 

Of the minor suggestions, I would consider changing "sclerotic" to "scleral" ring as it is more accurate to the anatomy (though both are understood to be the same thing).

Page 10 features the passage:

"Additionally, a sinus with relevance for the flight was found rostroventrally to the eye, corresponding to the infraorbital sinus"

This appears to refer to Jaensch's 2015 study on the lower respiratory tract of birds. Jaensch mentions this in passing without much detail beyond this. I recommend the authors rephrase this bit to say "may be relevant to flight" and then cite Jaensch 2015. Alternatively, they can expand a bit more on why this is relevant to flight.

Jaensch, S. (2015). Inspirational evolution: the avian lower respiratory tract.

Author Response

Dear Reviewer,

We sincerely thank the reviewer for their positive and encouraging feedback. We greatly appreciate the recognition of our work, particularly regarding the imaging methodology, discussion of study limitations, and the descriptive value of the figures. We are also pleased that the reviewer considers this study to be useful for veterinarians working with Charadriiformes species.

Comment 1: Of the minor suggestions, I would consider changing "sclerotic" to "scleral" ring as it is more accurate to the anatomy (though both are understood to be the same thing).

Response: We thank the reviewer for this valuable suggestion. We have replaced the term “sclerotic ring” with “scleral ring” throughout the manuscript to ensure anatomical accuracy and consistency. 

Comment 2: Page 10 features the passage: "Additionally, a sinus with relevance for the flight was found rostroventrally to the eye, corresponding to the infraorbital sinus". This appears to refer to Jaensch's 2015 study on the lower respiratory tract of birds. Jaensch mentions this in passing without much detail beyond this. I recommend the authors rephrase this bit to say "may be relevant to flight" and then cite Jaensch 2015. Alternatively, they can expand a bit more on why this is relevant to flight.

Response:  We thank the reviewer for this valuable comment and fully agree that the original phrasing could be interpreted as too conclusive. Following the suggestion, we have reworded the sentence to provide a more cautious description and added a paragraph in the Discussion section, incorporating the corresponding reference.

Reviewer 4 Report

Comments and Suggestions for Authors

In this study, a detailed anatomical and radiological characterization of the nasal cavity and sinuses of the yellow-legged gull (Larus michahellis atlantis) was performed through a combination of computed tomography and anatomical cross-section, demonstrating the practicality of CT imaging in the study of bird anatomy and helping to diagnose nasal pathology and structural abnormalities in this bird.

  1. The beginning of the abstract should clearly state the background and significance of conducting this research.
  2. The description of methods and results in the abstract is rather general; it is recommended to provide more specific details regarding the techniques used and the main findings.
  3. The introduction could be improved by specifically outlining the advances of CT technology in avian anatomy or the diagnosis of related pathologies.
  4. It is recommended to standardize the number of decimal places in the data presented in Table 1.
  5. The reference formatting should be consistent; please carefully check the reference list again.

Author Response

Dear Reviewer,

We would like to sincerely thank the reviewer for their thoughtful and constructive evaluation of our manuscript. We greatly appreciate the recognition of our work, which provides a detailed anatomical and radiological characterization of the nasal cavity and sinuses of the yellow-legged gull (Larus michahellis atlantis) through a combination of computed tomography (CT) and anatomical cross-sections. We are pleased that the reviewer acknowledges the practicality of CT imaging for studying avian anatomy and its relevance for diagnosing nasal pathologies and structural abnormalities in this species. The responses to the different comments are listed below.

Comment 1: The beginning of the abstract should clearly state the background and significance of conducting this research.

Response:
We thank the reviewer for this helpful suggestion. In the revised version of the manuscript, we have modified the beginning of the Abstract to provide clearer background information and emphasize the significance of conducting this research. The updated opening now contextualizes the relevance of studying the nasal cavity and associated sinuses in the yellow-legged gull, highlighting their anatomical and clinical importance in avian medicine and comparative anatomy.

Comment 2: The description of methods and results in the abstract is rather general; it is recommended to provide more specific details regarding the techniques used and the main findings.

Response:
We appreciate the reviewer’s valuable observation. In the revised version of the manuscript, the Abstract has been modified to include more specific methodological and result details. It now specifies that computed tomography (CT) was performed using a 16-slice helical scanner with both bone and pulmonary window settings, and that anatomical cross-sections were obtained from eight cadaveric specimens for correlation. Additionally, the main findings have been expanded to indicate that CT imaging effectively delineated key nasal structures such as the rostral, middle, and caudal nasal conchae, the infraorbital sinus, and associated cranial bones. These changes make the Abstract more informative and reflective of the study’s methodology and principal outcomes.

Comment 3: The introduction could be improved by specifically outlining the advances of CT technology in avian anatomy or the diagnosis of related pathologies.

Response:
We thank the reviewer for this valuable suggestion. In the revised version of the manuscript, we have expanded the Introduction to include a more specific overview of the advances in computed tomography (CT) technology and its applications in avian anatomy and pathology diagnosis. The new text highlights how CT has evolved as a non-invasive imaging modality that provides high-resolution, multiplanar, and three-dimensional visualization of complex cranial structures in birds, demonstrating the diagnostic and anatomical value of CT in avian species.

Comment 4: It is recommended to standardize the number of decimal places in the data presented in Table 1.

Response:
We thank the reviewer for this helpful comment. In the revised version of the manuscript, the numerical data in Table 1 have been standardized to one decimal place for all variables (weight, head length, and beak length) to ensure a uniform and professional presentation, while maintaining the original measurement accuracy.

Comment 5: The reference formatting should be consistent; please carefully check the reference list again.

Response:
We appreciate the reviewer’s observation. The entire reference list has been carefully reviewed and reformatted according to the Animals journal guidelines to ensure consistency in citation style, punctuation, and order of elements (authors, year, title, journal, volume, and pages).

Reviewer 5 Report

Comments and Suggestions for Authors

Overall Assessment:

This is a well-conducted and valuable anatomical study that successfully combines computed tomography (CT) and gross anatomical sections to describe the nasal cavity of yellow-legged gull. The research addresses a clear gap in the literature, as detailed anatomical data on the upper respiratory system of seabirds, particularly gulls, is scarce. The methodology is sound and appropriate for the objectives. The figures are of high quality and effectively support the text, providing an excellent visual atlas. The manuscript is generally well-written and structured. The findings are significant and will serve as a useful reference for veterinarians, rehabilitation specialists, and researchers in avian biology and medicine. 

Major Strengths:

  1. Novelty and Relevance: The study provides the first detailed, cross- sectional anatomical description of the nasal cavity in this species and subspecies, which is highly relevant given the gull's ecological importance and susceptibility to respiratory pathologists.
  2. Multimodal Approach: The combination of CT imaging (with different widow stings and 3D reconstructions) and physical anatomical sections is strong methodology that allows for excellent correlation and validation of findings. 
  3. Educational Value: The figures are exe perfect and constitute a high-quality anatomical atlas. The detailed labeling and side by side presentation of sections and CT scans are particularly useful for teaching and clinical reference.
  4. Discussion Context: The discussion effectively places the findings within the broader context of avian anatomy, noting similarities and differences with other bird species.

Specific Comments: 

  1. Introduction:

The introduction effectively establishes the ecological context and the need for the study. It would be beneficial to briefly mention the specific lack of data on gull nasal anatomy earlier on, not just seabirds in general, to further sharpen the research gap being filled.

  1. Materials and methods:

Section 2.2 (CT technique): The protocol is well described. Please clarify if the "pulmonary window" (WW=1400; ML=500) is also standardly referred to as a "soft tissue window" or "wide window" in other literature. A brief justification for choosing these specific window settings for highlighting nasal structures would be a minor improvement.

Section 2.3 (Anatomical sections) The 0.8 cm section thickness is quite large compared to the 0.6 mm CT slices. While this is acknowledged later in the limitations, it should be briefly noted here that this discrepancy means the anatomical sections are representative but not perfectly matched 1:1 with specific CT slices.

  1. Results:

The results are clear and descriptive. The flow rostral to caudal is logical.

Figures 2& 3: The text states these are paramedian sections, but the figure captions and results refer to them as such without specifying if they are sagittal or parasagittal. Please specify the plane (e.g., parasagittal section) in the captions for consistency with the transverse sections.

Figure 13: The caption refers to "subfigures A, B" as 3D reconstructions and "C, D" Osirix mask reconstructions. The difference between these two types of reconstructions could be very briefly explained in the caption for the non-specialist reader (e.g., " Volume-rendered 3D reconstruction (A, B) and semi-transparent 3D "mask" reconstruction isolating bone and air spaces (C, D) ").

  1. Discussion is strong. The comparison of the tripartite conchae to other species is excellent.

The paragraph on the infraorbital sinus is particularly good, discussing its clinical significance and contrasting findings with other studies.

Consider adding a sentence to speculate on the functional implication of the observed anatomy. For example, how might the large, spiraled middle concha and the extensive infraorbital sinus be adaptations for a marine, high-salt environment? While speculative, it would engage the reader in the potential biological significance pure morphology.

  1. Limitations:

The limitations section is commendably thorough and covers all major points (freeze-thaw artifacts, section thickness mismatch, lack of histology, functional data, and scanner variability). It is perfectly adequate as is.

Minor Corrections and slips:

Abstract: "vet-erinarians, biologists and researchers" please add spaces for readability.

Figure 12 Caption: "Tridimensional" is usually written as "Three-dimensional (3D) ".

Conclusion:

This manuscript presents a thorough and professionally executed anatomical study. The minor revisions suggested above are aimed at further enhancing clarity and context. The manuscript meets the standards for publication in Animals after these minor points are addressed. I commend the authors on their valuable contribution to avian morphological science.

Author Response

Dear Reviewer,

We sincerely thank the reviewer for their positive and encouraging evaluation of our work. We greatly appreciate their recognition of the study’s methodological soundness, the quality of the figures, and the relevance of the findings for veterinary and avian anatomical research. Our goal was precisely to fill the current anatomical knowledge gap regarding the upper respiratory system of seabirds, particularly gulls, and we are pleased that the reviewer found the study valuable in this regard. The response to the different comments are listed below.

Comment 1: Introduction:

The introduction effectively establishes the ecological context and the need for the study. It would be beneficial to briefly mention the specific lack of data on gull nasal anatomy earlier on, not just seabirds in general, to further sharpen the research gap being filled.

Response:

We thank the reviewer for this insightful suggestion. In the revised version of the Introduction, we have incorporated an explicit reference to the scarcity of anatomical data specifically concerning the nasal cavity of gulls, not only seabirds in general. This clarification has been added in the first part of the Introduction to better highlight the specific knowledge gap that this study addresses.

Comment 2a. Materials and methods:

Section 2.2 (CT technique): The protocol is well described. Please clarify if the "pulmonary window" (WW=1400; ML=500) is also standardly referred to as a "soft tissue window" or "wide window" in other literature. A brief justification for choosing these specific window settings for highlighting nasal structures would be a minor improvement.

Response:

In the revised version, we have clarified that the “pulmonary window” (WW = 1400; WL = –500) is not identical to the standard soft tissue window but was intentionally selected for its ability to enhance contrast between air-filled spaces and the surrounding thin bony and soft tissue structures. This setting has been used in previous CT studies of avian and small animal nasal anatomy [20, 24, 26, 27].

A brief justification has been added in Section 2.2, explaining that this window provided optimal delineation of the nasal cavity, conchae, and infraorbital sinus, which are characterized by complex air–bone interfaces.

Comment 2b. Materials and methods:

Section 2.3 (Anatomical sections) The 0.8 cm section thickness is quite large compared to the 0.6 mm CT slices. While this is acknowledged later in the limitations, it should be briefly noted here that this discrepancy means the anatomical sections are representative but not perfectly matched 1:1 with specific CT slices.

Response:
We thank the reviewer for this insightful observation. We fully agree that the 0.8 cm thickness of the anatomical sections, compared with the 0.6 mm CT slice thickness, may result in minor discrepancies between both imaging modalities. In the revised version, we have added a clarification in Section 2.3 noting that the anatomical sections are representative and intended for comparative orientation rather than exact 1:1 correspondence with individual CT slices.

Comment 3a. Results:

Figures 2& 3: The text states these are paramedian sections, but the figure captions and results refer to them as such without specifying if they are sagittal or parasagittal. Please specify the plane (e.g., parasagittal section) in the captions for consistency with the transverse sections.

Response:

We agree that the terminology should be consistent throughout the manuscript. In the revised version, we have specified the plane of section in the captions of Figures 2 and 3, clarifying that both correspond to parasagittal sections. This correction ensures consistency with the descriptions provided in the text and other figure captions.

Comment 3b. Results: Figure 13: The caption refers to "subfigures A, B" as 3D reconstructions and "C, D" Osirix mask reconstructions. The difference between these two types of reconstructions could be very briefly explained in the caption for the non-specialist reader (e.g., " Volume-rendered 3D reconstruction (A, B) and semi-transparent 3D "mask" reconstruction isolating bone and air spaces (C, D) ").

Response:

We thank the reviewer for this thoughtful suggestion. In the revised version of the manuscript, the caption of Figure 13 has been expanded to briefly clarify the difference between the two reconstruction types for non-specialist readers.

Comment 4. Discussion: Consider adding a sentence to speculate on the functional implication of the observed anatomy. For example, how might the large, spiraled middle concha and the extensive infraorbital sinus be adaptations for a marine, high-salt environment? While speculative, it would engage the reader in the potential biological significance pure morphology.

Response:
We thank the reviewer for this insightful and constructive suggestion. In the revised version, we have added a short paragraph in the Discussion section speculating on the possible functional implications of the observed anatomical features. Specifically, we discuss how the large, spiraled middle concha and the extensive infraorbital sinus may represent adaptive mechanisms related to the humid, saline marine environment inhabited by the yellow-legged gull. These structures could contribute to optimizing air filtration, thermoregulation, and moisture exchange, while also facilitating efficient salt elimination and respiratory conditioning. This addition provides a broader biological context to the morphological findings, as recommended.

Comment : Abstract: "vet-erinarians, biologists and researchers" please add spaces for readability.

Response:
We thank the reviewer for noticing this typographical issue. The spacing error in the phrase “vet-erinarians, biologists and researchers” has been corrected in the revised abstract. Furthermore, as suggested by another reviewer, the abstract has been comprehensively revised and reformatted to include a clearer background, more specific methodological details, and a concise summary of the main findings, thereby improving both clarity and readability.

Comment : Figure 12 Caption: "Tridimensional" is usually written as "Three-dimensional (3D) ".

Response:
We thank the reviewer for this helpful suggestion. The term “Tridimensional” in the caption of Figure 12 has been replaced with “Three-dimensional (3D)” in the revised version of the manuscript for consistency with standard scientific terminology.

Finally, we sincerely thank the reviewer for their positive and encouraging assessment of our work. We greatly appreciate their recognition of the scientific and anatomical value of this study, as well as their constructive minor suggestions, which have been carefully implemented to further improve the manuscript. We are grateful for their supportive comments and for considering our work suitable for publication in Animals.

Round 2

Reviewer 2 Report

Comments and Suggestions for Authors

Thank you very much for discussing the problems of the manuscript.

Reviewer 5 Report

Comments and Suggestions for Authors

Accept